# The Role of Resilience in the Relationship between Sociodemographic, Clinical Characteristics, and Social Support among Breast Cancer Patients in Serbia

**DOI:** 10.3390/healthcare11243184

**Published:** 2023-12-16

**Authors:** Sanja D Tomić, Goran Malenković, Armin Šljivo, Ermina Mujičić, Slobodan Tomić

**Affiliations:** 1Faculty of Medicine, University of Novi Sad, 21000 Novi Sad, Serbia; goran.malenkovic@mf.uns.ac.rs (G.M.); 907002d21@mf.uns.ac.rs (S.T.); 2Clinical Center, University of Sarajevo, 71000 Sarajevo, Bosnia and Herzegovina; sljivo95@windowslive.com (A.Š.); ermina.mujicic@fzs.unsa.ba (E.M.)

**Keywords:** breast cancer, Serbia, resilience, affective well-being, support

## Abstract

Background. The management of breast cancer treatments within the limitations of family, social, and professional life is emotionally burdening and negatively affects physical, psychological, and social well-being, reducing the overall quality of life of patients and their families. Methods: This cross-sectional descriptive–analytical study was conducted from March to August 2023 at the “Dr. Radivoj Simonović” General Hospital in Sombor. A total of 236 breast cancer patients participated in this study. The research was conducted using the following instruments: a questionnaire on sociodemographic and clinical characteristics of patients, the Berlin Social-Support Scales—for assessing social support—and the Connor–Davidson Resilience Scale—for assessing resilience. This study aimed to determine the predictors and levels of social support and resilience of breast cancer patients. We also wanted to examine whether resilience is a mediator between patients’ sociodemographic and clinical characteristics and levels of social support. Results: The total average value of social support was 3.51 ± 0.63, while on the resilience scale, the respondents achieved a total average score of 52.2 ± 9.63. Perceived and actually received social support of breast cancer patients were positively correlated with resilience [*p* < 0.01], while no statistically significant correlations were found for the need for support and satisfaction. The sets of predictors can significantly predict their effects on all types of perceived social support (emotional social support: 9%; perceived instrumental social support: 9%) and all types of received social support (actually received emotional social support: 8%; actually received instrumental social support: 7%; actually received informational social support: 8%). There is a potential mediating role of resilience in relation to sociodemographic factors, clinical characteristics, and the need for support. Conclusion: This study confirms that a strong connection exists between social support and resilience. However, the analysis did not confirm the mediating role of resilience between the sociodemographic and clinical characteristics on the one hand and social support on the other.

## 1. Introduction

The incidence and prevalence of breast cancer are constantly increasing; women all over the world most often face this particular diagnosis [1,2]. However, the number of breast cancer survivors is also increasing [3]. Coping with breast cancer, medical treatment protocols, the experience and life within the family, social, and professional contexts with all the limitations imposed by the disease, and its treatment can be upsetting and hard for patients and their families. The challenges that patients face involve a full range of negative effects on physical, psychological, and social functioning, resulting in the reduced quality of these patients’ lives [4,5,6,7]; this has encouraged many authors to search for factors that influence life after treatment [8,9,10,11]. As one of those factors, social support has been discussed as a valuable resource that mitigates the effects of stressful life events on health. It has consistently been associated with better mental health, self-esteem, physical health, and longevity [12].

Social support definitions state that it refers to the quality of supportive interactions a person has with other individuals and can play a significant role in well-being [13]. Social support is thought to function as a buffer to protect individuals from the physical and mental effects of stress [14]. As defined by the National Cancer Institute (NCI), social support is a network of family members, friends, neighbors, and community members who are available in times of need to provide psychological, physical, and financial assistance to cancer patients [15]. Cohen et al. state that adequate social support should meet the needs of patients and enable the development of an optimal method of coping with the disease because excessive support in a person’s life can negatively impact their activity and lead to the loss of independence [12].

There are several theoretical frameworks that explain the multidimensional effects of social support on the well-being and health of adults. The most widely accepted are the buffering and direct effect models. The first suggests that support reduces the harm of stressful events by preventing the individual from considering the situation as threatening or demanding [14]. On the other hand, the direct effect model suggests that social support is beneficial regardless of the amount of stress an individual perceives. In this model, the understanding that others are willing to help increases self-esteem and provides the individual with a sense of control over their situation [12].

Belonging to a social network can directly influence treatment outcomes by positively influencing treatment adherence and disease management [8]. An individual’s social network is essential to fully participate in and benefit from all facets of cancer treatment, including symptom management, care coordination, assistance with daily activities, and emotional support [16]. Despite the lack of consensus regarding the definition of social support, most authors emphasize the importance of perceived and actually received support [16,17,18,19]. Perceived social support refers to the expected availability of support in the future, whereas actual social support refers to the actual experiences of individuals. Furthermore, according to the type of support provided, it is possible to distinguish between instrumental, informational, and emotional support [12]. Emotional support is the care and support that inspires trust and a sense of belonging and love. Instrumental support is helping with practical tasks, such as transportation, childcare, and financial assistance. Informational support consists of knowledge, information, and advice [18]. For the purposes of this study, the above-mentioned concept was applied. These definitions take into account important aspects of this concept, i.e., the level of support received from diverse sources and the degree of satisfaction of individuals with this support, thus increasing resilience and giving individuals the courage to face challenges, improving their own adaptability and quality of life.

There is a general agreement among researchers that the disease usually limits patients’ participation in social activities, implicating that the opportunities to interact with others and their access to social support may be reduced [11,12,13,14]. Alternatively, the patients themselves may choose to withdraw from their social network. In both cases, this is related to a patient’s cancer experience, which depends on variables such as demographic (age, gender, socioeconomic status, etc.) and clinical (site of malignancy, stage of disease, and type of treatment) characteristics [11,15]. The results confirm that effective ways of coping with stress are significantly lower in women who have completed primary school and who have not undergone surgery. The levels of effective coping with stress in women subsides with increasing age, and as the perceived score of social support increases, so does the level of effective coping with stress. Findings show that social support and age significantly predict effective stress management [20]. In addition, negative relationships are found between the level of resilience and the time of diagnosis. Regarding the different treatments administered to the patients, experience with chemotherapy is the factor that produced the greatest impact in terms of increasing the level of resilience [21]. Previous studies indicate that one-quarter to one-third of breast cancer patients will develop anxiety and/or depression at some point during their treatment [20], and those under 50 are particularly likely to report psychological distress [22,23]. Psychological resistance, or resilience, can be defined as an individual’s ability to maintain or restore relatively stable psychological and physical functioning during or after significant stressful life events [24]. In cancer patients, resilience refers to a dynamic process that promotes successful adaptation to cancer-related hardships [25]. The relationship between resilience and social support in the cancer survivor population has already been established in some studies [26]. An Indian study shows that being optimistic not only attracts others but also encourages the establishment of social support networks. Moreover, optimism and social support play a vital role in improving the general well-being of cancer patients. Also, belonging to social networks has a direct impact on the treatment outcomes and is effective regarding the perceived vs. actually received social support and considering their distinct types, such as emotional, instrumental, and informational. Although a cancer diagnosis involves personal suffering, many women with breast cancer can develop the ability to resist and accept life’s crises, resulting in greater resilience.

Therefore, this study aimed to determine the levels of social support and resilience of breast cancer patients. We also wanted to examine the effects of predictors on the levels of social support and whether resilience was a mediator between patients’ sociodemographic and clinical characteristics and levels of social support.

## 2. Materials and Methods

This descriptive–analytical cross-sectional study was conducted from March to August 2023. The sample consisted of patients at the ‘Dr. Radivoj Simonović’ General Hospital in Sombor. All adult patients diagnosed with breast cancer took part in the research. Inclusion criteria were (1) diagnosis of breast cancer in women from stage 0 to stage IV; (2) completed cancer treatment, i.e., chemotherapy, hormone therapy, targeted therapy, immunotherapy, treatment of bone metastases, or any combination of these; and (3) the ability to communicate in the Serbian language. Participation in this study was voluntary and anonymous, with previously signed written consent from the participants. Exclusion criteria were any psychiatric or addictive disorders.

### 2.1. Research Instruments

#### Three Instruments Were Used in the Planned Research

##### Berlin Social-Support Scales [BSSS]

This is a set of self-report questionnaires developed by Schultz and Schwarzer [18] to assess social support. The BSSS includes six individual domains: perceived available support (8 items), need for support (4 items), support seeking (5 items), actually received social support (14 items), protective buffering (6 items), and satisfaction (1 item). The total number of BSSS items is 38. Using a multidimensional approach to measurement in the domain of perceived social support, it is possible to distinguish between two types of perceived social support, namely perceived emotional social support and perceived instrumental social support. Also, in the domain of actually received social support, three types are distinguished: actually received emotional social support, actually received instrumental social support, and actually received informational social support. The scale was validated on the sample of adult cancer patients with Cronbach’s alpha coefficients (a) ranging from 0.75 to 0.96 [18]. The response format for participants was consistent across domains. Participants rated their agreement with the statements on the BSSS scale as follows: strongly disagree (1), somewhat disagree (2), somewhat agree (3), and strongly agree (4). A higher score reflects a higher level of social support.

##### Connor–Davidson Resilience Scale [CD-RISC-25]

The CD-RISC-25 consists of statements that describe several aspects of resilience. The scale includes 25 items that measure resilience in 6 subscales: hardiness, coping, adaptability/flexibility, meaningfulness, optimism, and self-efficacy. Each item is scored from 0 to 4. A total score is obtained by summing all 25 items, giving a score that can range from 0 to 100. Lower scores indicate less resilience, and higher scores indicate greater resilience.

##### Sociodemographic Questionnaire and Clinical Characteristics of the Participants

The sociodemographic questionnaire for collecting data relevant to this study includes questions regarding the following: age, gender, educational attainment, employment, place of residence, socioeconomic status, and partner status. Socioeconomic status was assessed based on participants’ income. In relation to the amount of monthly income, they are classified into three categories. Average socioeconomic status is defined by income from EUR 600 to 1000; below the average is anything less than EUR 600, and above the average is anything greater than EUR 1000.

The clinical variables questionnaire to collect data relevant to this study included questions regarding the following: the time passed since cancer treatment, treatment modalities, fatigue, pain, and the presence of stress in the past year. Fatigue is defined as a feeling of physical exhaustion and lack of energy, and it is measured numerically by assessing its intensity at the time of the survey. Participants were asked to indicate the number on the NRS that best represented their current level of fatigue (“How tired are you feeling right now?”) using a 10-point fatigue scale (0 = “no fatigue”; 10 = “worst possible fatigue”). A 10-point fatigue scale has been well-validated to assess fatigue in people with cancer [27,28]. For the purposes of this study, fatigue was classified as follows: grades 0–3—no fatigue; 4–6—mild fatigue; and 7–10—intense fatigue. A visual analog scale was used to assess pain at the time of the survey. It was a 10 cm long numerical scale divided into 10 parts with three verbal descriptors. No scale was used to reassess stress, but participants were asked to clarify whether or not they had experienced stress during the last year.

### 2.2. Statistical Data Processing

Statistical analysis was performed using SPSS Statistics software (IBM SPSS Statistics for Windows, Version 22.0, Armonk, NY, USA). The results were tabulated, showing descriptive statistics data (frequencies and percentages for categorical data, as well as arithmetic means and standard deviations for quantitative data). The impact of patients’ sociodemographic and clinical characteristics on certain types of social support was calculated using the *t*-test and one-way analysis of variance (ANOVA). Correlations of social support and resilience were calculated using the Pearson correlation coefficient. Predictive variables consisted of sociodemographic characteristics in the first step, clinical characteristics in the second step, and dimensions of resilience in the third step of the hierarchical regression analysis. Considering significant predictor variables and their effects, a series of multiple regression analyses were conducted with social support (perceived and actual) as outcome variables. Cronbach’s Alpha reliability coefficient was used to assess the reliability of the instruments used.

### 2.3. Ethical Consent

In addition to the consent of the authors of the questionnaire, the consent of the Ethics Committee of the Faculty of Medicine in Novi Sad (Decision no. 01-39/34/1/2023) and the consent of the Ethics Committee of the General Hospital “Dr. Radivoj Simonović” in Sombor were obtained for conducting the research and using the research instruments (Decision No: 23-3108/2023).

## 3. Results

A total of 236 female participants took part in this study, with the highest percentage of women over 51 years old (65.7%). The majority (55.1%) completed primary education. About half of the participants (50.8%) were unemployed, and 61% lived in urban areas. In addition, 71.2% reported having a partner, and 50.8% rated their socioeconomic status as average. The treatment for the majority of participants was completed in the last 3 years (64.4%), and in the last year, half of them stated that they had no stress. Combined treatment, which includes both local and systemic methods, was administered to 67.8% of study participants. Almost half of the study participants (49.2% vs. 45.8%) experienced mild pain and fatigue. All other data regarding age, education, professional activity, place of residence, socioeconomic status, partners, and clinical variables in patients with breast cancer are shown in Table 1.

### 3.1. Descriptive Statistics and Correlation Coefficients CD-RISC-25 and BSSS

Table 2 shows the average scores for CD-RISC-25 and BSSS for the entire sample. CD-RISC-25 ranges from 0 to 100, with an average total score of 52.2 (SD = 9.63). The average total score for BSSS was 3.51 (SD = 0.63) out of 4.

Considering the correlation of resilience and social support, significant correlations were recorded with the intensity ranging from 0.255 to 0.487, all of which were positive. These results show that higher resilience is directly connected with higher social support. Resilience does correlate with the domains ‘need for support’ and ‘satisfaction’, but these correlations are not statistically significant (Table 3).

### 3.2. Effects of Sociodemographic and Clinical Variables on the Types of Social Support

The results showed that participants older than 50 were more inclined to seek support from others and protect their environment from information and diagnoses related to diseases. Participants with lower educational qualifications had a greater need for support and a higher level of support seeking, while employed participants protected their social environment more from illness and diagnosis information (Table 4).

Regarding other sociodemographic variables, there were no statistically significant differences in BSSS domains based on whether they lived in a rural or urban area, had a partner, and had socioeconomic status. Participants who underwent systemic therapy showed higher levels of support seeking and higher satisfaction with support. Regarding pain, it was observed that participants who reported no pain had significantly higher perceived instrumental and emotional support compared to those with mild to severe pain. In the domain of actually received social support, pain-free patients generally experienced more actually received emotional, instrumental, and informational social support. The effects of fatigue showed that it exists, but only in the domain of perceived social support and that it did not affect the actual received social support. Patients without fatigue perceived significantly higher perceived emotional and perceived instrumental social support and were more inclined to seek support. No statistically significant differences were observed between the remaining two groups of patients categorized according to fatigue (*p* > 0.05). Participants without stress reported significantly lower support-seeking compared to those who were stressed (Table 4).

### 3.3. Social Support Predictors

Having taken into account the significant predictor variables and their effects in the earlier stage of the analysis, a set of multiple regression analyses were conducted with social support (perceived and actually received) as the dependent variable (Table 5).

By looking at the results of a set of multiple regression analyses, it can be concluded that sets of predictors can significantly predict their effects on perceived emotional social support (9%), perceived instrumental social support (9%), actual emotional support (8%), actual instrumental support (7%), and actual informational support (8%).

### 3.4. Resilience as a Mediator in the Relationship between Patients’ Sociodemographic and Clinical Characteristics and Social Support

In order to verify whether resilience is a mediator between sociodemographic and clinical characteristics of patients and social support, a hierarchical multiple regression analysis was conducted (Table 6, Table 7 and Table 8). The results of the hierarchical analysis are presented only for the types of social support where we have identified the mediating role of resilience. For domains of social support as dependent variables, sociodemographic characteristics were entered in the first step of the hierarchical multiple regression analysis, clinical characteristics in the second step, and resilience aspects (Connor–Davidson Resilience Scale CD-RISC-25) in the third step.

The results indicate that greater adaptability, better emotional–cognitive regulation, and a lower level of searching for meaning in patients potentially lead to a greater need for support at the expense of age, socioeconomic status, and level of education. In particular, the effects of these three sociodemographic variables on the need for support decrease when resilience dimensions are considered, but their effects remain significant. This model explains 25% of the variance in support seeking via the predictor variables.

A significant contribution to perceived instrumental support is only provided by the level of education. In this model, perceived instrumental support is determined by the set of predictor variables to the extent of 20%.

There is no mediating role of resilience in the relationship between the time elapsed since treatment and actual received emotional social support. However, after introducing resilience, the results indicate that actual received emotional social support will be provided to more resilient patients. In this way, 32% of actual received emotional social support is explained.

## 4. Discussion

Social support can be viewed as an interactive construct, an interpersonal transaction that occurs between those who need help and those who provide support [29].

The importance of social support in improving positive treatment outcomes for people with chronic diseases and conditions has been confirmed in previous research [11,16,19,23]. Studies have shown that cancer patients who have higher levels of social support have a better quality of life and lower mortality rates [30,31]. Social support has been identified as one of the key factors in the daily lives of cancer patients [32]. Therefore, one of the aims of our study was to determine the level of social support in patients with breast cancer. The results of our research indicate a relatively high level of received social support [3.51 ± 0.63], which is in accordance with the results of previous studies [18,33]. It is common knowledge that many patients with cancer or other chronic diseases resort to their own network of social support and use different methods of self-management when faced with stressful situations. The reason for this might be the traditional family structure, which is still commonly present in Serbia. Dedication of family members and relatives to each other and good relationships with neighbors, particularly in difficult circumstances, such as disease, can be a significant factor, potentially impacting the results of this study. A higher level of social support after a breast cancer diagnosis can be valuable for the survival of these women because it improves their coping skills and increases the availability of cancer-related information [34]. The effects of social support suggest that it mainly operates as a stress alleviation model, with the greatest and most reliable benefits of support under conditions of psychological stress when support is most needed [12]. Observing the sociodemographic and clinical data of patients can provide relevant information on risk and protective factors that influence the adjustment to cancer diagnosis [35]. The results of our study suggest that age and a lower level of education contribute to increasing the need for social support. In contrast, the level of social support remains the same regardless of the partnership relationship, socioeconomic status, employment, and whether someone lives in urban or rural areas. Older cancer patients are faced with multiple challenges that include multiple losses, such as loss of strength to accomplish some of their routine activities at home, practical tasks, or social contacts [34]. Even when they receive support, these older women still suffer because of the loss of functional independence [36,37,38]. Unlike our results, a study conducted in Poland found that the education level did not influence the need for support among the participants [39], whereas living in urban areas was a positive and independent predictor of social support among survivors in China [40]. Our study also showed a higher score in the domain of protective buffering support among the employed participants. They proved to be protective of their environment regarding the information about the disease and diagnosis. These findings suggest that the social roles that confirm self-worth within the family, friends, or at work are extremely important for female social support experience [41,42]. The results of previous studies indicate that in the breast cancer patient population, work increases self-worth, quality of life, sense of purpose, and social integration [41]. In addition, such findings may indicate the presence of stigma related to the diagnosis and breast cancer treatment. It is necessary to conduct further research to provide closer insight into the matter. Among clinical variables, significant predictors of social support were the elapsed time since the treatment, the type of treatment, pain, fatigue, and stress. For some cancer patients, the amount of support can change over time, which indicates that the level and the type of support should be monitored. Namely, if the period of time since the treatment is short, there is a higher need for support [43]. Our study confirmed that the longer the treatment lasted, the less support they felt. Regarding different therapeutic treatment methods, experiencing systemic therapy is the factor that mostly impacted the increased level of seeking support and the higher level of satisfaction with the support. These findings are in accordance with previous studies [42]. Fatigue, pain, and stress can dramatically affect the quality of life of breast cancer patients, making them too exhausted to participate in regular activities and social events. In our study, participants who were without pain, fatigue, and stress reported significantly higher levels of social support. These results support earlier findings that found that women who reported higher levels of social support also reported lower levels of fatigue and pain [44,45]. Women with breast cancer can report varying levels of pain interference independent of pain intensity. Psychosocial factors, such as social support, may impact patients’ levels of pain interference. One of the significant psychosocial factors connected to negative emotional reactions of breast cancer patients is their psychological resilience. It is not about one personal characteristic but the result of an interaction between multiple personal characteristics and environmental factors [12]. Recent studies emphasize that psychological resilience as a personal factor and social support as an environmental factor function as a buffer against stress and increase the quality of life by reducing emotional stress among cancer patients [46,47]. Studies have found that resilience can strongly predict the patients’ fatigue from treatment, and well-developed resilience can help patients reduce treatment-induced functional impairment and shorten recovery time [48,49]. Coping style in oncology has been proven as one of the central factors in modulating the different individual psychological reactions towards the disease, the quality of life after receiving a cancer diagnosis, and the response and adjustment to treatment. The results of the correlation analysis of resilience and social support show a strong positive relationship. Namely, the more resilient female participants were, the higher the level of social support they received. The obtained results completely correspond to previously published studies [50]. Highly resilient cancer patients can depend less on psychosocial support in stress management when compared to those with lower resilience [51]. It is clear that social support improves general well-being, minimizes the risk of psychological stress, and represents a key factor in increasing the sense of hope among patients who have been diagnosed with cancer [52]. However, in our study, resilience did not function as a mediator in patients’ total social support and sociodemographic and clinical characteristics. Its potential mediating role was only found in the domain of seeking support and actually receiving emotional social support.

## 5. Conclusions

The obtained results serve as a fundamental basis for the development of a support system for breast cancer patients during and after treatment. Nurses should pay more attention to the resilience status and level of social support, as well as the coping style demonstrated by breast cancer patients. Methods and models of social support need to be adjusted to patients according to age and level of education. Identifying risk factors and inadequate coping mechanisms and creating social support programs targeting patients and their families so they can express their thoughts and feelings are vital. Further research is needed to identify factors that contribute to social support for breast cancer patients.

### Limitations of This Study

Our research faced several limitations. To begin with, it was a cross-sectional study, whereas a longitudinal approach would be more suitable for monitoring the changes in perceptions and needs for social support over time. Another limitation is related to the need to expand the group of participants, which could be achieved by introducing a comparative analysis with male breast cancer patients. Ultimately, we found a significant correlation between resilience and social support, but more research is needed to fully understand this relationship.

## Figures and Tables

**Table 1 healthcare-11-03184-t001:** Sociodemographic and clinical characteristics.

Sociodemographic Variables	%	N
Age (years)		
≤50	34.3	81
≥51	65.7	155
Education		
Other than college	55.1	130
College/University	44.9	106
Professional activity		
Active	49.2	116
Inactive	50.8	120
Residence		
Rural	39.0	92
Urban	61.0	144
Self—reported financial standing		
Below average	33.5	79
Average	50.8	120
Above average	15.7	37
Partner		
Yes	71.2	168
No	28.8	68
Clinical variables	%	N
Stress		
Yes	47.5	112
No	52.1	123
Time since treatment (years)		
≤1	22.9	54
>1–≤3	41.5	98
>3	35.6	84
Type of treatment		
Only local therapy	25.4	60
Systemic therapy	6.8	16
Combined therapy	67.8	160
The pain		
Without pain	38.1	90
Mild	49.2	116
Strong	12.7	30
The fatigue		
Non-existent	17.8	42
Mild	45.8	108
Intense	36.4	86

**Table 2 healthcare-11-03184-t002:** Average scores on CD-RISC-25 and BSSS scales.

	Total Range Min–Max	Total Mean (SD)
CD-RISC-25	0–100	52.2 (9.63)
BSSS	0–4	3.51 (0.63)

**Table 3 healthcare-11-03184-t003:** Correlation of social support and resilience.

	PESS	PISS	NS	SS	SATIS	ARES	ARInsS	ARInfS	CD-RISC-25
PESS	1								
PISS	0.535 **	1							
NS	0.262 **	0.356 **	1						
SS	0.129	0.373 **	0.490 **	1					
SATIS	0.052	0.059	0.047	−0.005	1				
ARES	0.293 **	0.406 **	0.173	0.275 **	0.145	1			
ARInsS	0.465 **	0.469 **	0.294 **	0.262 **	0.123	0.436 **	1		
ARInfS	0.331 **	0.344 **	0.247 **	0.302 **	0.127	0.371 **	0.374 **	1	
CD-RISC-25	0.487 **	0.469 **	0.182	0.255 **	0.106	0.371 **	0.418 **	0.337 **	1

** *p* < 0.01. PESS—perceived emotional social support; PISS—perceived instrumental social support; NS—need of support; SS—seeking support; SATIS—satisfaction; ARES—actual received emotional social support; ARInsS—actual received instrumental social support; ARInfS—actual received information social support.

**Table 4 healthcare-11-03184-t004:** Effects of sociodemographic and clinical variables on types of social support.

Types of Social Support	Sociodemographic Variables	Mean ± SD	Statistic Test/*p*
Seeking support	Age (years)		
	≤50 years	2.65 ± 0.83	
	≥51 years	2.95 ± 0.66	t =−2.34/0.03
	Education		
	Primary/High school	2.91 ± 0.78	
	Bachelor	2.60 ± 0.75	T = 2.24/0.03
Protective buffering support	Age (years)		
	≤50	2.49 ± 0.76	t = −2.05/0.05
	≥51	2.71 ± 0.65	
	Professional activity		
	Active	2.78 ± 0.78	
	Inactive	2.39 ± 0.69	t = 2.08/0.04
Need for support	Education		
	Primary/High school	2.95 ± 0.62	t = 2.68/0.01
	Bachelor	2.64 ± 0.64	
	Clinical variables	Mean ± SD	Statistic test/*p*
Need for support	Type of treatment		
	Only local therapy	2.62 ± 0.92	
	Systemic therapy	3.20 ± 0.51	F = 3.78/0.03
	Combined therapy	2.73 ± 0.75	
	Stress		
	Yes	3.03 ± 1.02	t = 1.94/0.05
	No	2.71 ± 0.82	
	Fatigue		
	Non-existent	3.04 ± 0.99	
	Mild	2.32 ± 0.68	F = 6.90/0.00
	Intense	2.89 ± 0.80	
Satisfaction	Pain		
	Without the pain	3.67 ± 0.99	
	Mild	3.16 ± 0.68	F = 7.53/0.00
	Strong	2.75 ± 0.80	
	Type of treatment		
	Only local therapy	3.04 ± 0.99	
	Systemic therapy	3.60 ± 0.68	F = 3.33/0.04
	Combined therapy	3.46 ± 0.80	
Perceived instrumental social support	Time since treatment (years)		
	≤1	3.48 ± 0.71	
	>1–≤3	3.01 ± 0.90	F = 3.17/0.05
	>3	3.12 ± 0.77	
	Pain		
	Without the pain	3.38 ± 0.93	
	Mild	3.08 ± 0.52	F = 3.59/0.03
	Strong	2.56 ± 0.80	
	Fatigue		
	Non-existent	3.58 ± 0.93	
	Mild	2.81 ± 0.52	F = 5.19/0.01
	Intense	3.26 ± 0.80	
Perceived emotional social support	Pain		
	Without the pain	3.45 ± 0.49	
	Mild	3.06 ± 0.78	F = 6.14/0.00
	Strong	2.81 ± 0.99	
	Fatigue		
	Non-existent	3.64 ± 0.36	
	Mild	3.00 ± 0.80	F = 3.90/0.02
	Intense	3.26 ± 0.67	
Actual received emotional social support	Pain		
	Without the pain	3.18 ± 0.41	
	Mild	2.90 ± 0.58	F = 5.62/0.01
	Strong	2.71 ± 0.63	
Actual received instrumental social support	Pain		
	Without the pain	3.55 ± 0.93	
	Mild	3.15 ± 0.52	F = 4.77/0.01
	Strong	2.92 ± 0.80	
Actual received information social support	Pain		
	Without the pain	3.34 ± 0.97	
	Mild	2.84 ± 0.77	F = 5.84/0.00
	Strong	2.87 ± 0.84	

**Table 5 healthcare-11-03184-t005:** Predictors of social support.

	PESS		PISS		ARES		ARInsS		ARInfS	
	F = 6.36,		F = 3.84,		F = 11.20,		F = 9.45,		F = 10.40,	
	df = 2123,		df = 3120,		df = 1124,		df = 1126,		df = 1127,	
	*p* < 0.001,		*p* < 0.005,		*p* < 0.001,		*p* < 0.001,		*p* < 0.001,	
	R^2^ = 0.09		R^2^ = 0.09		R^2^ = 0.08		R^2^ = 0.07		R^2^ = 0.08	
	β	t	β	t	β	t	β	t	β	t
Age	-	-	-	-	-	-	-	-	-	-
Education	-	-	-	-	-	-	-	-	-	-
Time since treatment	-	-	−0.19	−2.11 *	-	-	-	-	-	-
Pain	−0.29	−3.37 **	−0.24	−2.67 **	−0.29	−3.35 **	−0.26	−3.07 **	−0.28	−3.32 **
Fatigue	−0.06	−0.73	−0.03	−0.36						
Stress	-	-	-	-	-	-	-	-	-	-
Professional activity	-	-	-	-	-	-	-	-	-	-

** *p* < 0.01, * *p* < 0.05. PESS—perceived emotional social support; PISS—perceived instrumental social support; ARES—actual received emotional social support; ARInsS—actual received instrumental social support; ARInfS—actual received information social support.

**Table 6 healthcare-11-03184-t006:** Mediator effect of sociodemographic, clinical characteristics, and resilience to the expression of support needs.

	Step 1		Step 2		Step 3	
	F = 3.70,		F = 2.70, *p* = 0.015,		F = 2.77, *p* = 0.001,	
	*p* = 0.001,		R^2^ = 0.15, ΔR^2^ = 0.02,		R^2^ = 0.25, ΔR^2^ = 0.12,	
	R^2^ = 0.13		*p* = 0.767		*p* = 0.005	
	β	t	β	t	β	t
Age	−0.246	−2.579 *	−0.259	−2.225 *	−0.226	−2.709
Education	−0.436	−4.174 *	−0.450	−3.995 *	−0.428	−4.484 *
Professional activity	0.220	2.190 *	0.168	1.556	0.015	0.138
Residence	−0.046	−0.441	−0.071	−0.632	0.080	0.694
Self-reported financial standing	0.319	3.110 *	0.328	3.114 *	0.271	2.666
Partner	−0.029	−0.277	−0.055	−0.507	−0.092	−0.889
Stress	-	-	0.106	1.029	0.097	0.994
Time since treatment	-	-	−0.006	−0.063	−0.013	−0.136
Type of treatment	-	-	−0.085	−0.854	−0.065	−0.674
Pain	-	-	−0.072	−0.676	−0.029	−0.273
Fatigue	-	-	0.046	0.436	0.044	0.416
Hardiness	-	-	-	-	0.134	0.898
Coping	-	-	-	-	−0.164	−1.236
Adaptability	-	-	-	-	0.267	2.092 *
Meaningfulness	-	-	-	-	−0.371	−2.648 *
Optimism	-	-	-	-	0.138	1.331
Regulation of emotion and cognition	-	-	-	-	0.248	2.328 *
Self-efficacy	-	-	-	-	0.104	0.859

* *p* < 0.05.

**Table 7 healthcare-11-03184-t007:** Mediator effect of sociodemographic, clinical characteristics, and resilience to the expression of perceived instrumental support.

	Step 1		Step 2		Step 3	
	F = 1.83,		F = 1.61, *p* = 0.100,		F = 2.29, *p* = 0.010,	
	*p* = 0.090,		R^2^ = 0.07, ΔR^2^ = 0.02,		R^2^ = 0.20, ΔR^2^ = 0.13,	
	R^2^ = 0.05		*p* = 0.260		*p* = 0.006	
	β	t	β	t	β	t
Age	−0.084	−0.833	−0.073	−0.699	−0.118	−1.197
Education	−0.214	−1.922	−0.202	−1.811	−0.238	−2.275 *
Professional activity	0.235	2.166	0.181	1.601	0.067	0.616
Residence	−0.096	−0.860	−0.109	−0.938	−0.048	−0.431
Self-reported financial standing	0.257	2.356	0.228	2.094	0.196	1.867
Partner	0.128	1.120	0.112	0.970	0.062	0.561
Stress	-	-	0.015	0.137	0.069	0.674
Time since treatment	-	-	0.013	0.119	0.049	0.475
Type of treatment	-	-	0.078	0.752	−0.035	−0.348
Pain	-	-	−0.088	−0.792	0.047	0.410
Fatigue	-	-	−0.170	−1.540	−0.091	−0.831
Hardiness	-	-	-	-	0.105	0.672
Coping	-	-	-	-	0.132	0.925
Adaptability	-	-	-	-	0.240	1.762
Meaningfulness	-	-	-	-	−0.012	−0.080
Optimism	-	-	-	-	−0.053	−0.489
Regulation of emotion and cognition	-	-	-	-	−0.002	−0.015
Self-efficacy	-	-	-	-	0.146	1.168

* *p* < 0.05.

**Table 8 healthcare-11-03184-t008:** Mediator effect of sociodemographic, clinical characteristics, and resilience to the expression of actual received emotional social support.

	Step 1		Step 2		Step 3	
	F = 0.90,		F = 1.19, *p* = 0.300,		F = 1.97, *p* = 0.020,	
	*p* = 0.051,		R^2^ = 0.15, ΔR^2^ = 0.09,		R^2^ = 0.32, ΔR^2^ = 0.17,	
	R^2^ = 0.06		*p* = 0.192		*p* = 0.006	
	β	t	β	t	β	t
Age	−0.021	−0.193	0.007	0.066	−0.027	−0.265
Education	−0.084	−0.718	−0.033	−0.277	−0.070	−0.632
Professional activity	0.177	1.558	0.182	1.542	0.078	0.674
Residence	−0.189	−1.589	−0.235	−1.902	−0.199	−1.662
Self-reported financial standing	0.207	1.788	0.178	1.536	0.138	1.249
Partner	0.010	0.088	0.006	0.055	−0.047	−0.890
Stress	-	-	−0.111	−0.995	−0.200	−0.447
Time since treatment	-	-	−0.247	−2.265	−0.061	−1.901
Type of treatment	-	-	0.007	0.062	−0.589	−0.589
Pain	-	-	−0.041	−0.352	−0.023	−0.195
Fatigue	-	-	−0.141	−1.223	−0.014	−0.119
Hardiness	-	-	-	-	0.410	2.483
Coping	-	-	-	-	−0.018	−0.127
Adaptability	-	-	-	-	0.189	1.363
Meaningfulness	-	-	-	-	−0.060	−0.388
Optimism	-	-	-	-	0.053	0.465
Regulation of emotion and cognition	-	-	-	-	−0.142	−1.236
Self-efficacy	-	-	-	-	0.001	0.006

## Data Availability

The data are available from the authors on personal request.

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
