# Peer review of "The Role of Resilience in the Relationship between Sociodemographic, Clinical Characteristics, and Social Support among Breast Cancer Patients in Serbia"

_healthcare, 2023, doi:10.3390/healthcare11243184_

Round 1
Reviewer 1 Report
Comments and Suggestions for Authors
The purpose of the article is interesting, however several points need to be strengthened before it can be considered for publication. The first thing is that the introduction is more aimed at giving definitions that show the state of the art on the subject and the need to review the role of resilience and social support in this population. Therefore, the purpose of the article is not entirely clear. Furthermore, in the methodology they do not explain in the sociodemographic data section how they measured stress, pain and how they made the classification according to economic status; It seems that they were the patients' perceptions and if so, the comparisons they make and the conclusions they reach are not valid. Regarding social support, they compare the subscales of the instrument, which I am not sure is appropriate given how the instrument was constructed. Due to the above, I consider that the construction of the article should be improved both theoretically and methodologically so that its conclusions are supported and it is of interest to the academic community and for the development of better tools for women with this disease.
Comments on the Quality of English LanguageIn general, from my point of view, English needs to be improved, since there are several words that are not very well used and confusing phrases.
Author Response
Dear Editor,
Thank you for giving us the opportunity to resubmit our manuscript titled The role of resilience in the relationship between sockodemographic and clinical characteristics among breast cancer patients in Serbia. We really appreciate the time and effort that you and the reviewers have dedicated to providing your valuable feedback on our manuscript. We are grateful to the reviewers for their insightful comments on our paper. We have tried to incorporate changes to reflect most of the suggestions provided by the reviewers.
Here is a point-by-point response to the reviewers’ comments and concerns.
All changes in the manuscript are marked in yellow.
Comments from reviewer 1:
|
Comment: |
Answer: |
1. |
The purpose of the article is interesting, however several points need to be strengthened before it can be considered for publication. The first thing is that the introduction is more aimed at giving definitions that show the state of the art on the subject and the need to review the role of resilience and social support in this population. Therefore, the purpose of the article is not entirely clear. |
Thank you for recognising the importance of our study.We have expanded and clarified the aim of the study. |
2. |
Furthermore, in the methodology they do not explain in the sociodemographic data section how they measured stress, pain and how they made the classification according to economic status; It seems that they were the patients perceptions and if so, the comparisons they make and the conclusions they reach are not valid. |
Thank you for your comment.We have corrected all that were suggested. |
3 |
Regarding social support, they compare the subscales of the instrument, which I am not sure is appropriate given how the instrument was constructed. Due to the above, I consider that the construction of the article should be improved both theoretically and methodologically so that its conclusions are supported and it is of interest to the academic community and for the development of better tools for women with this disease. |
Thank you for your comment.We have corrected all that were suggested. Poređenja subskala smo iybrisali. |
Reviewer 2 Report
Comments and Suggestions for Authors
The manuscript describes a study on the relationship between resilience and social support in breast cancer patients. In principle, the topic is of interest for the readers of “Healthcare”.
The sample size is large enough. The instrument chosen for measuring social supportis not well suited for answering the research questions, and the statistical calculations are not appropriate.
Abstract:
Methods: It would be good to report the assessment instruments.
Results: The description of the study sample (education, treatment) is not so relevant here; the degree of social support and the relationship between social support and resilience is more important.
Conclusion: The conclusion (family support) is not justified by the results. In the total results section, family support is not mentioned at all.
The English is good. However, there are several superficialities, especially in the tables, e.g.
Table 3:
Lokally therapy => Local therapy (or, more exactly: Only local therapy, since the patients with combined therapy are not included here)
Combinet therapy => Combined therapy
Time since treatment: please include the time frame: months? years?
“Lokally th,” => Local therapy only
Table 4:
Stres => Stress
Koping => Coping
purpose => Purpose
Selfefficancy => Self-efficacy
The instrument for recording social support (BSSS) is confusing and, in my opinion, therefore unsuitable. It comprises five dimensions and a satisfaction scale.
In addition, two of the five dimensions are further broken down into two or three sub-dimensions, so that we end up with 11 dimensions, some of which contain each other.
Introduction:
The Introduction gives a very good theoretical insight in the research field. However, the authors do not report specific results that have already been obtained in the field of their own study, the association between resilience and social support.
Methods:
The description of the instrument BSSS in point 2.2.1 is not optimal. It would be better to precisely name the five domains:
line 127: The authors write “perceived emotional and instrumental support”, and one does not know whether this is one dimension or whether these are two dimensions.
line 140: “The overall support scores can be obtained by summing up… for each domain.” However, the authors reported that only one dimension, actual support, is composed of these three components (emotional, instrumental, informational support).
Please write the total number of items of the BSSS.
line 144: A detailed description of which items belong to which scales is not necessary.
A description of the assessment of stress, pain, and fatigue is totally missing.
line 161: the sentence “The results are presented in tabular form” is not necessary.
Results:
line 173: The statements are not correct. “… the largest percentage being women older than 51 years who had completed elementary education.” This means a combination of these two characteristics, but they should be treated separately.
“while 61 % of them lived in a city with a partner (71.2 %)”. Once more, this is a combination of two characteristics, and this is not correct.
“… had been completed within the last 3 years (41.5 %)”. This is also incorrect, since 22.9 % + 41.5 % completed their treatment within the last 3 years.
Please write the mean age of the sample.
The headings of the tables should be shortened. It is not necessary to write in each table that the breast cancer patients were from Serbia.
The authors are not consistent in writing the names of the BSSS scales.
actual received support – support actually received – real social support
emotional support for life activity (Tab 3). Is this identical with actual support: emotional subdimension?
Due to the confusing abundance of dimensions of the BSSS, an exact, uniform terminology is particularly important.
In Table 2 the authors report mean scores of the dimensions and compare them one with another. In the verbal description (line 202), they also compare sub-dimensions (e.g. emotional actual support) without writing the underlying mean scores.
I think that the comparison of these mean scores is of limited value since the questionnaire was not designed for comparing such sub-dimensions.
In Table 3 the authors report associations between sociodemographic and clinical values on the one side and BSSS results on the other. The only report statistically significant results. When I see it correctly, they performed 11 (6 BSSS scales plus 5 BSSS subscales) * 11 (6 sociodemographic variables + 5 clinical variables) = 121 comparisons; 18 of these were statistically significant. It would be clearer to report the results of all comparisons, the significant and the non-significant ones. If this is too much, the authors may consider restricting to the most relevant of the 11 scales of the BSSS.
line 240: “with no stress reported … higher seeking of support…” This is not true: Patients with no stress report only 2.71, while those with stress report 3.03.
l. 248: CDRIS-25 => CD-RISC-25
Please report the mean scores of the CD-RISC scales.
Tables 4,5 and 6:
For 3 out of the 11 scales of the BSSS the authors report analyses of the relationship between resilience and social support. First, it is unclear why these 3 scales were chosen.
When the authors intend to describe the additional explained variance explained by the resilience variables, they should also report R² of step 2 and compare this R² with R² of step 3.
“p=0.00” is not sufficient, please write 3 decimal points.
R2 is not correct, please write R².
I think that in Table 4 the effect of education in step 3 (t = -4.484) is also statistically significant.
A better understanding of the relationship between resilience and social support could be reached if the authors would write a table with correlations between resilience scales and BSSS scales.
Discussion
At the beginning, the authors repeat a description of their sample, this is not necessary.
l. 286: “a large amount of perceived and actual support”. This would be justified if the authors had compared their results with normative scores or mean scores of other examinations. So it remains totally unclear whether the mean scores are high or low.
l. 304: “support in practical tasks such as transportation, childcare, financial assistance.” In the Results section I do not see results that support this statement.
l. 313: “from members of their immediate environment and family.” This is also not shown in the results section.
What is completely missing in the Discussion is the aspect of resilience, though this is a main aim of the study and also written in the title of the manuscript.
Comments on the Quality of English LanguageThe English is good; however, there are several superficialities in the tables.
Author Response
Dear Editor,
Thank you for giving us the opportunity to resubmit our manuscript titled The role of resilience in the relationship between sockodemographic and clinical characteristics among breast cancer patients in Serbia. We really appreciate the time and effort that you and the reviewers have dedicated to providing your valuable feedback on our manuscript. We are grateful to the reviewers for their insightful comments on our paper. We have tried to incorporate changes to reflect most of the suggestions provided by the reviewers.
Here is a point-by-point response to the reviewers’ comments and concerns.
All changes in the manuscript are marked in yellow.
Comments from reviewer 2:
|
Comment: |
Answer: |
1 |
The manuscript describes a study on the relationship between resilience and social support in breast cancer patients. In principle, the topic is of interest for the readers of “Healthcare”. The sample size is large enough. The instrument chosen for measuring social supportis not well suited for answering the research questions, and the statistical calculations are not appropriate. |
Thank you for your comment and recognizing the importance of research topic. Dear Sir/Madam, thank you for pointing out the omissions. In accordance with your suggestions, we have corrected the research objectives and statistical calculations |
2 |
Abstract: Methods: It would be good to report the assessment instruments. |
We have added this in the Abstract. |
3 |
Results: The description of the study sample (education, treatment) is not so relevant here; the degree of social support and the relationship between social support and resilience is more important. |
We have modified the abstract according to your suggestions. |
4 |
Conclusion: The conclusion (family support) is not justified by the results. In the total results section, family support is not mentioned at all. |
Corrected. |
5 |
The English is good. However, there are several superficialities, especially in the tables, e.g. Table 3: Lokally therapy = Local therapy (or, more exactly: Only local therapy, since the patients with combined therapy are not included here) Combinet therapy = Combined therapy Time since treatment: please include the time frame: months? years? “Lokally th,” =Local therapy only Table 4: Stres =Stress Koping = Coping purpose = Purpose Selfefficancy =Self-efficacy |
We thank the reviewer for this comment. The language mistakes are corrected by English proficient proof reader. |
6 |
The instrument for recording social support (BSSS) is confusing and, in my opinion, therefore unsuitable. It comprises five dimensions and a satisfaction scale. In addition, two of the five dimensions are further broken down into two or three sub-dimensions, so that we end up with 11 dimensions, some of which contain each other. |
Thank you for your comment. |
7 |
Introduction: The Introduction gives a very good theoretical insight in the research field. However, the authors do not report specific results that have already been obtained in the field of their own study, the association between resilience and social support. |
Thank you for your comment.We have added a study which explains the connection of this phenomena. |
8 |
Methods: The description of the instrument BSSS in point 2.2.1 is not optimal. It would be better to precisely name the five domains: line 127: The authors write “perceived emotional and instrumental support”, and one does not know whether this is one dimension or whether these are two dimensions. |
Thank you for your comment. We have corrected it. |
9 |
line 140: “The overall support scores can be obtained by summing up… for each domain.” However, the authors reported that only one dimension, actual support, is composed of these three components (emotional, instrumental, informational support). Please write the total number of items of the BSSS.
|
Thank you for your comment. We have corrected it. |
10 |
line 144: A detailed description of which items belong to which scales is not necessary. |
Deleted. |
11 |
A description of the assessment of stress, pain, and fatigue is totally missing. |
Thank you for your comment. We have added |
12 |
line 161: the sentence “The results are presented in tabular form” is not necessary. |
We have removed it. |
13 |
Results: line 173: The statements are not correct. “… the largest percentage being women older than 51 years who had completed elementary education.” This means a combination of these two characteristics, but they should be treated separately. |
Thank you for your comment. We have corrected it. |
14 |
“while 61 % of them lived in a city with a partner (71.2 %)”. Once more, this is a combination of two characteristics, and this is not correct. “… had been completed within the last 3 years (41.5 %)”. This is also incorrect, since 22.9 % + 41.5 % completed their treatment within the last 3 years. |
Thank you for your comment. We have corrected it. |
15 |
Please write the mean age of the sample. |
Mean age couldn’t be provided as we asked our respondents just to declare their age group. |
16 |
The headings of the tables should be shortened. It is not necessary to write in each table that the breast cancer patients were from Serbia. |
Thank you for your comment. We have corrected it. |
17 |
The authors are not consistent in writing the names of the BSSS scales. actual received support – support actually received – real social support |
Thank you for your comment. We have corrected it. |
18 |
Emotional support for life activity (Tab 3). Is this identical with actual support: emotional subdimension? |
Thank you for your comment, yes we corrected it. |
19 |
Due to the confusing abundance of dimensions of the BSSS, an exact, uniform terminology is particularly important. |
Thank you for your comment. We have corrected it. |
20 |
In Table 2 the authors report mean scores of the dimensions and compare them one with another. In the verbal description (line 202), they also compare sub-dimensions (e.g. emotional actual support) without writing the underlying mean scores. I think that the comparison of these mean scores is of limited value since the questionnaire was not designed for comparing such sub-dimensions. In Table 3 the authors report associations between sociodemographic and clinical values on the one side and BSSS results on the other. The only report statistically significant results. When I see it correctly, they performed 11 (6 BSSS scales plus 5 BSSS subscales) * 11 (6 sociodemographic variables + 5 clinical variables) = 121 comparisons; 18 of these were statistically significant. It would be clearer to report the results of all comparisons, the significant and the non-significant ones. If this is too much, the authors may consider restricting to the most relevant of the 11 scales of the BSSS. |
Thank you for your comment. We have corrected it.
Due to the large number of variables, we decided to present only the significant predictor variables. Therefore, we conducted a set of multiple regression analyses and displayed them in a new table." |
21 |
line 240: “with no stress reported … higher seeking of support…” This is not true: Patients with no stress report only 2.71, while those with stress report 3.03. |
Thank you for your comment. We have corrected it. |
22 |
l. 248: CDRIS-25 = CD-RISC-25 |
Thank you for your comment. We have corrected it. |
23 |
Please report the mean scores of the CD-RISC scales. |
Added. |
24 |
Tables 4,5 and 6: For 3 out of the 11 scales of the BSSS the authors report analyses of the relationship between resilience and social support. First, it is unclear why these 3 scales were chosen. |
Thank you for your comment. The mentioned scales are presented because statistically significant differences were observed on them, indicating the potential mediating role of resilience between the sociodemographic and clinical characteristics of patients and social support. Mediation was not observed on the remaining scales of the BSSS questionnaire; therefore, the results of the hierarchical analysis are not displayed. |
25 |
When the authors intend to describe the additional explained variance explained by the resilience variables, they should also report R² of step 2 and compare this R² with R² of step 3. “p=0.00” is not sufficient, please write 3 decimal points. R2 is not correct, please write R².
|
Thank you for your comment. We have corrected it. |
26 |
I think that in Table 4 the effect of education in step 3 (t = -4.484) is also statistically significant. |
Thank you for your comment. We have corrected it. |
27 |
A better understanding of the relationship between resilience and social support could be reached if the authors would write a table with correlations between resilience scales and BSSS scales. |
Thank you for your comment. Dodali smo table sa korelacijama |
28 |
Discussion At the beginning, the authors repeat a description of their sample, this is not necessary. |
Thank you for your comment. We have corrected it. |
29 |
l. 286: “a large amount of perceived and actual support”. This would be justified if the authors had compared their results with normative scores or mean scores of other examinations. So it remains totally unclear whether the mean scores are high or low. |
Thank you for your comment. We have corrected it. |
30 |
l. 304: “support in practical tasks such as transportation, childcare, financial assistance.” In the Results section I do not see results that support this statement. |
Thank you for your comment. We have corrected it. |
31 |
l. 313: “from members of their immediate environment and family.” This is also not shown in the results section. |
Thank you for your comment. We have corrected it. |
32 |
What is completely missing in the Discussion is the aspect of resilience, though this is a main aim of the study and also written in the title of the manuscript. |
We added |
Round 2
Reviewer 1 Report
Comments and Suggestions for Authors
I thank the authors of the article for all the changes made, from my perspective it is accepted in its current version.
Author Response
Dear Reviewer,
thank You very much for such kind words, comments and suggestions for the revision of our paper.
Reviewer 2 Report
Comments and Suggestions for Authors
The authors have made many additions and changes. After the first version already contained a number of formal errors, the newly submitted form again contains many formal inconsistencies. This makes it difficult to evaluate the content. It is not the task of a reviewer to point out formal errors, but to assess the content and point out possible improvements. However, this is difficult to do with a sloppy and flawed manuscript. I therefore ask the authors to submit a formally flawless manuscript before I can evaluate its content. Examples of formal inconsistencies are:
Table 1: Locally therapy
Table 2: It is unclear what is meant by "Items by Subscales". There are no items in the table.
Selfefficancy => Self-efficacy (remove the hyphen and the “n” before the y)
Total scor => Total score
It is unclear why "Total score” is in the middle and not at the left part of the table.
Table 3: The abbreviations are sometimes with a dot (eg., SS., SATIS.) and sometimes not, this is inconsistent.
Table 4: Sociodemografic = Sociodemographic
Age in the first part without "years", in the second part with "years"
"Local therapy,"
…
The subheadings have completely different formats and are therefore not clearly recognizable as subheadings. Four subheadings appear in four different styles:
Descriptive statistics and correlation coefficients SD Risc-25 and BSSS: small font; is this really a heading?
Examining the effects... beginning at the left margin, ending with a dot.
Predictors of social support… not in bold, too much left
Mediation of Resilience... : This is written in italics, and most words are written in capitals.
There are also inconsistencies in the References: Journal names sometimes in italics and sometimes not, sometimes dots after the abbreviated words and sometimes not.
The changes made are often linguistically poor or incorrect. Example in the abstract: "using Berlin Social-Support Scales, Connor-Davidson Resilience Scale and Sociodemographic-Questionnaire and Participant's Clinical Characteristics."
Within this single sentence, there are already lots of mistakes.
Please note that the inconsistencies given here are only examples. It is therefore not sufficient to correct only these. Please provide a formally correct form. As the changes you made in the current form are often linguistically poor, please also have all these parts of the text linguistically revised by a native speaker.
Comments on the Quality of English LanguageThe English is not acceptable, especially the newly written sentences.
Author Response
Dear Reviewer,
thank You very much for comments and suggestions for the paper. We are sorry for such a inconvenience that was done to accelerate the publication of this paper. We have corrected English, made tables and everything more concise and readable for the audience. Thank You very much.

Round 3
Reviewer 2 Report
Comments and Suggestions for Authors
I still do not consider the BSSS instrument used to be suitable for conducting such studies. However, now that the authors have made it this far, the paper may be published once the remaining errors have been corrected.
The authors have removed a large number of errors from the last version. However, some inconsistencies remain:
end of abstract: "... was not determied". This is unclear: did the authors not analyze the relationship, or did they analyze it and failed to detect the mediation role?
p. 7, line 221 (SD=9.63) (SD=0.63). The second statement is incorrect here.
p. 7, l. 220: "Resilience does not correlate..." This not correct: Resilience does correlate, but it does not statistically significantly correlate.
l. 235: It would be a little bit more concise to remove "Investigating the" from the heading.
Sometimes capital letters are used in the table headings (Tab 4) and sometimes not (Tab 3).
P. 8, center: 2.78=0.78 is incorrect
P. 10, line 263 “…with social support (…) as variables”. Probably meant: ... as dependent variable.
p. 11, line 76: Table = > Tables
Table 6:
In step 2, new independent variables are added. Therefore, I do not believe that R² decreases from 0.15 to 0.13. It should increase, as with Table 7. Perhaps the authors have mixed up the values for R² from step 1 and step 2. In addition, step 2 says: R2 R²; that is nonsense.
It would be better to either consistently write a zero in front of the decimal point for delta R² or not, but not to switch.
Table 7: The figures here are consistent; the differences in R² can be derived from the individual values for R².
p. 13 and p. 14: Self-efficancy = > self-efficacy (this is disappointing: I corrected it already two times.)
References. There are still discrepancies here:
sometimes a comma after the name (e.g.Ref. 1), sometimes not.
No consistent bold print of the years (e.g. Ref. 8)
Partially missing spaces between names (e.g. Ref. 7)
Journal names: Sometimes dots after words (e.g. Ref 13), sometimes not
Ref. 26: First letter of Indian is not in italics
Ref. 32: : 2016, 1;2(9). what does this mean? Are there two entries for the volume numbers?
similar to Ref. 34
Ref. 40: 12;22(1) does not make sense either. The authors may have forgotten to delete the 12, which comes from the date.
Author Response
Dear Editor,
Thank you for giving us the opportunity to resubmit our manuscript titled The role of resilience in the relationship between sockodemographic and clinical characteristics among breast cancer patients in Serbia. We really appreciate the time and effort that you and the reviewers have dedicated to providing your valuable feedback on our manuscript. We are grateful to the reviewers for their insightful comments on our paper. We have tried to incorporate changes to reflect most of the suggestions provided by the reviewers.
Here is a point-by-point response to the reviewers’ comments and concerns.
All changes in the manuscript are marked in red.
Comments from reviewer 2:
|
Comment: |
Answer: |
1. |
The authors have removed a large number of errors from the last version. However, some inconsistencies remain: end of abstract: "... was not determied". This is unclear: did the authors not analyze the relationship, or did they analyze it and failed to detect the mediation role? p. 7, line 221 (SD=9.63) (SD=0.63). The second statement is incorrect here. p. 7, l. 220: "Resilience does not correlate..." This not correct: Resilience does correlate, but it does not statistically significantly correlate. l. 235: It would be a little bit more concise to remove "Investigating the" from the heading. Sometimes capital letters are used in the table headings (Tab 4) and sometimes not (Tab 3). P. 8, center: 2.78=0.78 is incorrect P. 10, line 263 “…with social support (…) as variables”. Probably meant: ... as dependent variable. p. 11, line 76: Table = > Tables Table 6: In step 2, new independent variables are added. Therefore, I do not believe that R² decreases from 0.15 to 0.13. It should increase, as with Table 7. Perhaps the authors have mixed up the values for R² from step 1 and step 2. In addition, step 2 says: R2 R²; that is nonsense. It would be better to either consistently write a zero in front of the decimal point for delta R² or not, but not to switch. Table 7: The figures here are consistent; the differences in R² can be derived from the individual values for R². p. 13 and p. 14: Self-efficancy = > self-efficacy (this is disappointing: I corrected it already two times.) References. There are still discrepancies here: sometimes a comma after the name (e.g.Ref. 1), sometimes not. No consistent bold print of the years (e.g. Ref. 8) Partially missing spaces between names (e.g. Ref. 7) Journal names: Sometimes dots after words (e.g. Ref 13), sometimes not Ref. 26: First letter of Indian is not in italics Ref. 32: : 2016, 1;2(9). what does this mean? Are there two entries for the volume numbers? similar to Ref. 34 Ref. 40: 12;22(1) does not make sense either. The authors may have forgotten to delete the 12, which comes from the date.
|
Thank you for your comment..
We have corrected.
We have corrected.
We have corrected the according to your suggestions.
We have corrected.
We have corrected Table 4.
We have corrected.
We have corrected.
We have corrected.
Table 6: We have corrected.
Table 6, 7, 8: We have corrected.
We have corrected.
We have corrected.
We have corrected.
We have corrected.
We have corrected.
We have corrected. We have corrected. We have corrected. We have corrected
|